# Vitamin D Status and Seasonal Variation among Danish Children and Adults: A Descriptive Study

**DOI:** 10.3390/nu10111801

**Published:** 2018-11-20

**Authors:** Louise Hansen, Anne Tjønneland, Brian Køster, Christine Brot, Rikke Andersen, Arieh S. Cohen, Kirsten Frederiksen, Anja Olsen

**Affiliations:** 1Diet, Genes and Environment, Danish Cancer Society Research Center, DK-2100 Copenhagen, Denmark; annet@cancer.dk (A.T.); anja@cancer.dk (A.O.); 2Department of Prevention and Information, Danish Cancer Society, DK-2100 Copenhagen, Denmark; brk@cancer.dk; 3Danish Health Authority, Department of Prevention, DK-2300 Copenhagen, Denmark; chb@sst.dk; 4National Food Institute, Research Group for Risk-Benefit, Technical University of Denmark, DK-2800 Lyngby, Denmark; rian@food.dtu.dk; 5Statens Serum Institute, Department of Congenital Diseases, Center for Newborn Screening, DK-2300 Copenhagen, Denmark; aco@ssi.dk; 6Statistics and Pharmacoepidemiology, Danish Cancer Society Research Center, DK-2100 Copenhagen, Denmark; kirstenf@cancer.dk

**Keywords:** vitamin D status, vitamin D deficiency, 25(OH)D, seasonal variation, StatusD, Denmark

## Abstract

The aim of the present study was to describe vitamin D status and seasonal variation in the general Danish population. In this study, 3092 persons aged 2 to 69 years (2565 adults, 527 children) had blood drawn twice (spring and autumn) between 2012 and 2014. A sub-sample of participants had blood samples taken monthly over a year. Serum 25-hydroxyvitamin D (25(OH)D) concentrations were measured by liquid chromatography mass spectrometry, and information on supplement use was assessed from questionnaires. Seasonal variations in 25(OH)D concentrations were evaluated graphically and descriptively, and status according to age, sex, and supplement use was described. It was found that 86% of both adults and children were vitamin D-sufficient in either spring and or/autumn; however, many had a spring concentration below 50 nmol/L. A wide range of 25(OH)D concentrations were found in spring and autumn, with very low and very high values in both seasons. Among adults, women in general had higher median 25(OH)D concentrations than men. Furthermore, vitamin D supplement use was substantial and affected the median concentrations markedly, more so during spring than autumn. Seasonal variation was thus found to be substantial, and bi-seasonal measurements are vital in order to capture the sizable fluctuations in vitamin D status in this Nordic population.

## 1. Introduction

Vitamin D insufficiency and its implications has been a topic of intense debate among health professionals for many years; in fact, low vitamin D status in relation to major non-communicable diseases has been investigated in numerous observational studies [1,2,3]. However, while adequate concentrations of vitamin D are indisputably important for optimal bone health, potential associations between vitamin D deficiency and disease remain disputed and the results are in general conflicting [4,5]. 

As vitamin D is produced in the skin in response to solar radiation, low vitamin D status is thought to be common in Denmark and other Nordic countries during winter due to inadequate sun exposure from October to March [6,7]. The number of vitamin D status measurements performed on indication or otherwise has therefore increased dramatically in many Western countries in recent years [8,9]; in Denmark, some health professionals have postulated that Danes in general have low vitamin D status. However, this has not been measured in a systematic manner among the general population previously and is thus an area where data is lacking.

Recommendations for preventive vitamin D supplementation in Denmark already exist for some sub-groups considered to be at risk of low vitamin D status [10]; this includes children under 2 years, elderly persons aged 70 years and above, dark-skinned individuals or persons who wear protective clothing, those who live in nursing homes, and those at risk of osteoporosis. However, due to the lack of systematic data on the general population status, national policies for vitamin D recommendations are difficult to formulate. Several studies describing vitamin D status in Denmark have been published, but these were either based on blood samples drawn from persons visiting their general practitioner [11] or on sub-groups, e.g., children, women, or immigrants [12,13,14,15,16,17]. Many studies were furthermore based on single blood sample measurements [11,13,14,15,16,17,18,19]; thus, if seasonal fluctuations are sizable, data on vitamin D status may be erroneous if only collected during either the high or low season.

In the current study, vitamin D status was therefore investigated among a sample of the general Danish population not advised by the Danish Health Authority to take vitamin D supplements. The study was denoted StatusD and included adults and children aged 2–69 years from three areas in Denmark and entailed blood samplings twice (spring and autumn) for all participants. The current paper describes seasonal variations among the participants stratified by sex, age, and supplement use as well as status assessments combining their spring and autumn measurements.

## 2. Methods 

### 2.1. Study Description

The StatusD study was established between 2012 and 2014, and 3408 persons between 2 and 69 years from the general population (recruited in three regions in Denmark: Copenhagen Central area (63%), Odense (19%), and Kolding (18%)) were enrolled using a web-based enrollment system. They were recruited through mailed letters sent to addresses that were obtained using the Danish Civil Registration System (CPR). In order to reach children as well as adults, we contacted both individual persons as well as families. Persons with dark skin, those who wore protective clothing or covering during the summer (e.g., due to religious reasons), those who were rarely outside or actively avoided sunlight, those who lived in nursing homes, and those with increased risk of developing osteoporosis were not eligible to participate. Pregnant women were also ineligible, as vitamin D supplements are recommended during pregnancy; however, if they became pregnant between the two samplings, they could still participate. 

The participants were invited to visit one of the three study centers twice, in spring and autumn, and all were instructed to complete an electronic lifestyle and dietary questionnaire concurrent with the time of blood sampling. Serum samples were taken in January–March (12% in January, 40% in February, 48% in March), constituting “spring” and in late August through October with a few samples taken in November (12% in late August, 42% in September, 44% in October, and 2% in November), constituting “autumn”. Post hoc assessment revealed that 61 persons did not fulfil the study inclusion criteria and were excluded, leaving a total of 3347 participants in StatusD. A more detailed description of the study recruitment (including participation rate) and blood samplings has been published previously [20]. The study was approved by the Regional Ethics Committee in Copenhagen (H-4-2012-068), and written informed consent was provided from all participants.

Of the 3347 participants, 232 persons were excluded, as they only contributed with one sample, thereby being ineligible in the assessment of seasonal variation, and a further 23 persons were excluded due to incomplete questionnaire or measurement data. A total of 255 persons were therefore excluded, leaving a study population of 3092 persons, encompassing 527 children (<18 years) and 2565 adults (≥18 years), defined by age at first sampling.

To assess seasonal variation in detail, all adult participants recruited in the spring sampling in 2013 were asked if they would also contribute a monthly blood sample for the following 11 months; 167 persons (40% men, 60% women) agreed to do so, constituting a sub-population of monthly participants. Blood samples were taken on the first Monday and Tuesday of every month from April 2013 to February 2014. The median number of samples per person was 11 (minimum 4; maximum 12) with the highest number of samples per month found in October (*n* = 162) and the minimum in August (*n* = 133); 93% of the monthly participants provided eight samples or more.

### 2.2. Measurement of 25-hydroxyvitamin D (25(OH)D) in Serum Samples

The serum sample from each participant was collected in Hemogard plain red-top tubes (Becton Dickinson, Albertslund, Denmark) and stored at −80 °C until use. Analysis of 25-hydroxyvitamin D3 and D2 was performed at Statens Serum Institute (Copenhagen, Denmark) using liquid chromatography-tandem mass spectrometry (LC-MSMS) in positive mode using the “MSMS vitamin D” kit from Perkin Elmer (Waltham, MA, USA). The two forms in combination denote 25-hydroxyvitamin D (25(OH)D). All samples were measured by single determination with a total coefficient of variation below 15%. The Statens Serum Institute participates in DEQAS (the Vitamin D External Quality Assessment program).

For the majority of the participants, 25(OH)D2 was not measured in significant concentrations. It was found 53% of the participants had no measurable 25(OH)D2 concentration. For the remaining participants, the distribution was as follows: 25% had concentrations where 25(OH)D2 equaled less than 1% of the total; 55% had 25(OH)D2 concentrations between 1% and 5% of the total 25(OH)D; and for the remaining 20% that had a measurable 25(OH)D2 concentration, the proportion ranged from 5% to 25% of the participants’ total. However, there were seven participants who had higher concentrations than that (from 29% to 73% of total 25(OH)D). More details on the analysis method and standardization have been previously described [20].

### 2.3. Questionnaire Information on Supplement Use

The questionnaires contained five questions on supplement use (including name and vitamin D content) as well as a frequency question, and vitamin D intake from supplements in μg per day was calculated using this information. Persons with a daily supplement intake of less than 2 μg vitamin D per day were denoted as non-supplement users to avoid imposing a daily dose on infrequent/rare users; the cut-off value was chosen based on the intake distribution and a realistic assessment of minimum supplement dosages.

### 2.4. Descriptive Statistics

All descriptive analyses were performed using SAS Enterprise Guide 5.1 (SAS Institute Inc., Cary, NC, USA), and figures were created using SAS 9.4 (SAS Institute Inc., Cary, NC, USA) and Inkscape 0.91 (Free Software Foundation, Inc., Boston, MA, USA). Concentration ranges were calculated for men, women, boys and girls in relevant age groups and stratified by supplement use. The included age groups were 2–5 years (kindergarten), 6–14 years (school children), and 15–17 years (adolescents). Age ranges for adults ranged from 18 to 69 years in 10-year intervals. Descriptive results combining both spring and autumn concentrations were calculated among all, supplement users and among non-supplement users (the latter being those that did not use supplements in neither spring nor autumn). We considered 50 nmol/L as an appropriate cut-off for determining adequate 25(OH)D concentrations, as this is the consensus in Denmark [10]. As such, 25(OH)D concentrations below 25 nmol/L are termed to indicate vitamin D deficiency, whereas concentrations between 25–50 nmol/L indicate insufficiency. Finally, the difference in vitamin D concentrations (delta) by season for each person was calculated and summarized. The term vitamin D is used to describe overall status in the present paper, while 25(OH)D concentrations are reported for assay analyses.

## 3. Results

### 3.1. Overall Status Stratified by Sex and Age

Of the 255 participants that were excluded, 72 were children (28%), and the median age (5–95%) of these was 11 (2–17) years. In comparison, the 527 children who donated two samples make up 17% of the total number of included participants, and the median age (5–95%) was also 11 (3–17). The median age (5–95%) of the excluded adults was 41 (20–65) years compared to 47 (21–67) in the included participants. Concerning sex differences, 61% were women/girls among the excluded participants, which was a rate slightly higher than that of the included population (58% women/girls). Of the 255 who were excluded, 232 were excluded because they only contributed one blood sample; 127 contributed an autumn sample only and 105 a spring sample only. Median (5–95%) 25(OH)D concentrations were 70 (45–103) nmol/L and 50 (18–110) nmol/L for autumn and spring, respectively.

The median (5–95%) 25(OH)D concentration among the StatusD population overall was 62 (31–103 nmol/L) for adults (*n* = 2565) and 57 (30–88) nmol/L for children (*n* = 527). For adults in autumn, median concentrations were 72 (39–115) nmol/L, while in spring they were 50 (19–99) nmol/L. For children, the corresponding values were 67 (40–99) nmol/L in autumn and 44 (19–84) nmol/L in spring. The percentile distributions were similar for adults and children (results not shown in Table); however, adults had higher maximum concentrations (maximum in spring for adults: 183 nmol/L, for children: 114 nmol/L; maximum in autumn for adults: 171 nmol/L, for children: 132 nmol/L).

Table 1 shows the median concentrations of 25(OH)D stratified by season, age and supplement use among boys/men (*n* = 1312) and among girls/women (*n* = 1780). Women in general had higher median 25(OH)D concentrations than men, and supplement users consistently had higher median concentrations and ranges than non-supplement users with the difference seemingly greater in spring. Among women, 45% were supplement users in the spring, with fewer being so in autumn (38%). The same was true for men (supplement users in spring: 31%; autumn: 27%). Among boys and girls, the median concentrations and ranges differed slightly but with no apparent pattern. It was found that 33% of girls and 35% of boys were supplement users in the spring, and 32% and 25% were supplement users in autumn. The highest median concentrations were found in the highest and the lowest age groups for both sexes (2–6 years and 60+ years, respectively), but this pattern was not as noticeable when considering non-supplement users only.

Table 2 shows that among men, 14% were vitamin D-deficient (below 25 nmol/L 25(OH)D) in spring, but only 1% were vitamin D-deficient in autumn. Furthermore, 45% of men were vitamin D-insufficient (between 25 and 50 nmol/L 25(OH)D) in spring and 17% in autumn. A similar pattern is apparent for women; however, with lower proportions (9% of women were vitamin D-deficient in spring). For children, a similar distribution was found; 18% of boys and 9% of girls were vitamin D-deficient in spring, but only 0.4% and 0.8%, respectively, were so in autumn. With regard to vitamin D insufficiency, this was seen for 42% of boys in spring (dropping to 16% in autumn); a greater proportion of girls was vitamin D-insufficient in spring (50%), dropping to 12% in autumn. Very few (1% of both men and women) were severely vitamin D-deficient (<12.5 nmol/L 25(OH)D); almost none were severely vitamin D-deficient in autumn, but 7% of boys aged 15–17 years were vitamin D-deficient in the spring. 

### 3.2. Seasonal Variation of 25(OH)D Concentrations

In Figure 1, all 25(OH)D concentrations are plotted for all adults and children (all: black circles) and for the monthly participants (monthly: gray squares). Concentrations ranging widely from very low to very high were found in both spring and autumn months. The median (5–95 percentiles) delta seasonal difference among adults was 20 (2–54) nmol/L; when restricted to non-supplement users, the difference was slightly larger at 26 (5–58) nmol/L. Among children, results were similar; the median delta seasonal difference was 22 (2–51) nmol/L for all, and 27 (7–52) nmol/L when restricted to non-supplement users.

Overall, 86% of both adults and children had a 25(OH)D concentration above 50 nmol/L at some point during the year (nearly all had the highest concentration in autumn) (Table 3). When restricted to non-supplement users, 81% of both adults and children remained vitamin D-sufficient during either spring and/or autumn. Almost half (49%) of all adults were vitamin D-sufficient in both seasons, but when restricted to non-supplement users, this number was substantially lower; only 29% of adults had 25(OH)D concentrations above 50 nmol/L in both spring and autumn. Among children, 38% were vitamin D-sufficient all year, with this value dropping to 22% when restricted to non-supplement users. 

Figure 2 illustrates 25(OH)D concentrations among the 167 monthly participants with the lowest concentrations found in March and April and the highest in August, creating a sine curve. Also illustrated are the concentrations among supplement and non-supplement users, the latter being those who did not use supplements spring nor autumn (*n* = 77). The curves parallel each other with the one among non-supplement users slightly downward shifted, and the highest concentrations found among supplement users.

Among the 167 participants who agreed to donate a blood sample monthly, the distribution of educational length was investigated: 27% had a received a long education (4+ years, PhD), 35% had received a medium-length education (3–4 years), 14% had received a short education (<3 years), 14% had received vocational training, and 10% had received an elementary/high school-length education. In comparison, 20% of the main participants had a long education, while the remaining distribution was as follows: 31% had received a medium-length education, 14% a short education, 20% had received vocational training, and 16% had received an elementary/high school-length education. The median age (5–95%) of the monthly participants and those in the main study was 49 (22–67) years versus 46 (18–66) years, respectively. Finally, more women than men agreed to participate in the monthly blood sampling (72% women, 28% men) while the sex distribution among the main participants was 59% women, 41% men.

## 4. Discussion

Our study is the first systematic, descriptive overview of the vitamin D status (measured as 25(OH)D concentrations) among a sample of Caucasian Danish children and adults not advised to take vitamin D supplements by the Danish Health Authority. Several studies concerning vitamin D status have been performed in Denmark, but none in the general population, with both a spring and an autumn measurement including all ages from 2 to 69 years and among both men/boys and women/girls. Overall, we found considerable seasonal variation in 25(OH)D concentrations among children and adults in StatusD; concentration ranges were highly variable, with low and high concentrations found in all months. Large delta seasonal differences were seen; furthermore, women generally had higher concentrations than men, and the highest median concentrations were found in the oldest and youngest age groups. Supplement use was common among participants (especially women), ranging from 25 to 45% when considering all participants and affected the concentrations appreciably, more so during the spring than the autumn.

The median concentrations of the monthly measurements in Figure 2 show that maximum values were reached in early August when the sampling was conducted. However, the blood sampling period for the overall autumn samplings started in late August, meaning we may in fact have underestimated the highest concentrations for some. Minimum values were seen in February and March among all participants overall as well as among the monthly participants. Finally, more than half of both adults and children were vitamin D-insufficient during spring, but the majority of both adults and children reached vitamin D sufficiency during autumn, even when disregarding use of vitamin D supplements.

One of the major strengths of the study is that two measurements were available for 93% of the study population, and in order to assess status in both seasons, we only included these persons. This enabled us to assess 25(OH)D concentrations for all at both the presumed high and low point of the one’s vitamin D reserve period. Each participant provided a spring and an autumn blood sample and answered detailed questions regarding supplement use concurrently, meaning that we were able to assess the delta seasonal variation in 25(OH)D concentrations and to consider the impact of supplement use. We examined the characterization of the excluded participants and found that a larger proportion were children, and that among the excluded adults, they were slightly younger than those who were included in the analyses.

Furthermore, for 167 persons, we had up to 12 measurements over a year, making it possible to examine seasonal status in detail, showing a clear sine curve with highs in autumn and lows in winter/early spring. We also assessed the impact of supplement use on seasonal status graphically. Of those who agreed to participate, a higher proportion had a longer education than that of the main participants; they were also slightly older, and more women than men participated. 

We included participants from three different regions in Denmark to ensure geographic variation, and we included both children and adults in the study. However, the overall participation rate was low as described previously [20], which may impact the generalizability of the results. When comparing the StatusD population with the general Danish population in the same age range, we found that StatusD participants in general had higher educational status (comparisons not shown). Finally, vitamin D status assessed through 25(OH)D was measured using LC-MS, which is considered the optimal method for measuring vitamin D metabolites [21,22]. The samples in the current project were measured using a high quality kit from Perkin Elmer at a lab that participates in DEQAS, the international quality assessment scheme for vitamin D metabolites. However, as substantial heterogeneity in vitamin D concentrations has been reported with regard to different laboratories and analysis methods, the validity and comparability of 25(OH)D measurements should be considered when comparing studies on vitamin D [23,24].

Use of vitamin D supplements was not cause for exclusion in the study; rather, we stratified the results showing vitamin D status with and without supplement use. As expected, use of supplements was greater during spring, and the difference in median 25(OH)D concentrations between seasons was larger among non-supplement users. Supplement use was assessed meticulously, but we cannot rule out that some misclassification still exists; however, the errors are likely to be random rather than systematic.

As the sun is the primary driver of vitamin D synthesis in the body, it may seem surprising that 25(OH)D concentrations as high as 80–90 nmol/L were seen for the spring concentrations among non-supplement users. This is most likely due to either travelling to a sunny destination in the winter months (for which 14% of the population answered on their lifestyle questionnaires that they did) or possibly due to genetic differences. Recent research indicates that genetic predisposition plays an important role in the susceptibility to vitamin D insufficiency, and that both uptake of vitamin D through diet and supplements as well as skin-formation after solar exposure are highly dependent on individual genetic variations [25,26,27].

In the current study, 10% of the women and 15% of the men were vitamin D deficient in spring, which are similar values, albeit slightly higher, to the estimated 6–10% described in a previous report [28]. However, some previously published Danish studies have found a higher prevalence of vitamin insufficiency than the current study, but there are several methodological issues to consider. First, the previous claims of high prevalence of vitamin D insufficiency or even deficiency in Denmark stem from studies on at-risk sub-groups [13] or from only certain age segments [12,14,17,18,19]. In the Copenhagen vitamin D study, CopD, performed on 25(OH)D measurements from persons who visited their general practitioner, 54.4% were found to be vitamin D-insufficient [11]; however, it is conceivable that persons who visit their doctor might have a predisposing condition that leads to lower 25(OH)D concentrations. Second, estimation of circulating 25(OH)D concentrations is highly dependent on analysis method, and several studies used assays now considered to be less reliable [11,14]. Finally, delta seasonal variation has not been assessed previously in as large a population sample as in StatusD. Madsen et al. found in the Food with vitamin D study, VitmaD, that 9% of their Danish study population aged 4–60 years (*n* = 755) were vitamin D-insufficient; however, this was based on singular blood samples taken only in late summer (with the conceivable maximum values), indicating again the need for repeated measurements [18]. To our knowledge the only study with repeated measurements was one performed on Danish adolescent girls and elderly women (sampled twice in winter and once in summer). The majority (87–93%) of the girls had concentrations below 50 nmol/L during the winter season, while <20% had this in summer [12]. In comparison to this, we found that two-thirds of girls aged 6–14 years (66%) were vitamin D-insufficient or worse in spring, while only 14% were so in autumn.

We found that 80–90% of both adults and children had a 25(OH)D concentration of more than 50 nmol/L at some point during the year. If restricted to non-supplement users, still more than 80% of both adults and children had an adequate 25(OH)D concentration at one point. However, the opposite should also be noted: 51% of adults (71% if restricted to non-supplement users) had a 25(OH)D concentration below 50 nmol/L in at least one of the seasons. For children, the corresponding values were 62% and 78% among non-supplement users. It is as yet unknown whether low vitamin D status is only present when 25(OH)D concentrations are low all year round or if one low 25(OH)D concentration at some point during the year implies low status [29]. For this reason, it is impossible to say whether the finding that between half and three-quarters of the adult and child populations, respectively, had 25(OH)D concentrations below 50 nmol/L (primarily during winter) implies corresponding harmful effects to health. This needs to be investigated and clarified in further studies. Furthermore, percentages describing prevalence of low 25(OH)D concentrations are also subject to the arbitrariness of the cut-off for sufficiency, which is often debated among health professionals. The chosen cut-off of 50 nmol/L is based on bone health and may not be appropriate for assessment of general health outcomes. 

## 5. Conclusions

In this sample of the general Danish population, not including at-risk groups but encompassing children and adults between 2 and 69 years old, we found substantial seasonal variation in the 25(OH)D concentrations. Most participants were vitamin D-sufficient in autumn, but many experienced vitamin D insufficiency during the spring, emphasizing the need for individual, bi-seasonal measurements when assessing status.

## Figures and Tables

**Figure 1 nutrients-10-01801-f001:**
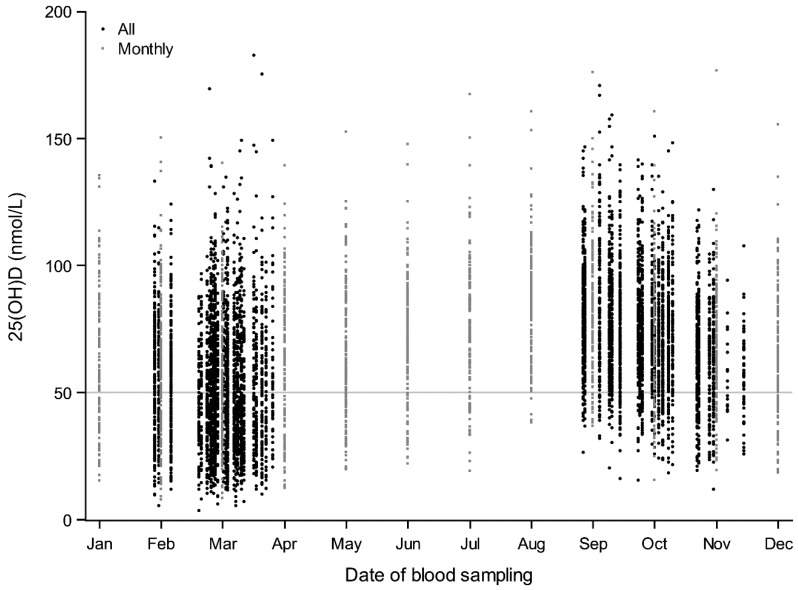
Seasonal variation in serum 25-hydroxyvitamin D (25(OH)D) concentrations among all participants (black dots; *n* = 3092) and only monthly participants (small gray squares; *n* = 167).

**Figure 2 nutrients-10-01801-f002:**
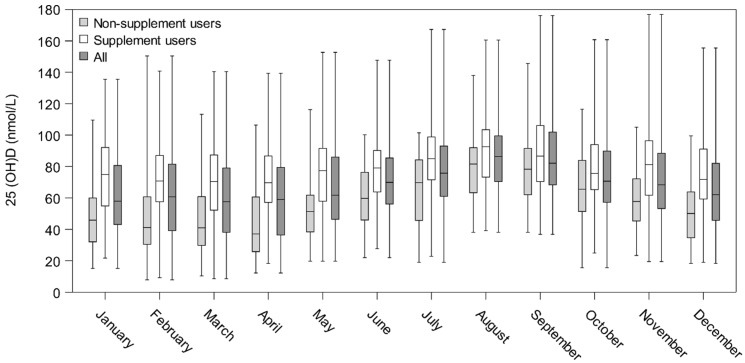
Box plot (box from Q1 to Q3) showing 25(OH)D (nmol/L) concentrations among all monthly participants (*n* = 167) and non-supplement users (*n* = 77). Tails show min and max values, while the line, –, indicates the median.

**Table 1 nutrients-10-01801-t001:** Median serum 25-hydroxyvitamin D (25(OH)D) concentrations (nmol/L) stratified by season, sex, and supplement use among children (2–17 years) and adults (18–69 years).

Age in Years by Sex (*n*)	Spring					Autumn				
	All	SU ^a^		NSU ^b^		All	SU ^a^		NSU ^b^	
	Median (5–95%)	*n*	Median (5–95%)	*n*	Median (5–95%)	Median (5–95%)	*n*	Median (5–95%)	*n*	Median (5–95%)
Boys										
2–5 (*n* = 36)	56 (24–98)	18	63 (25–107)	18	37 (24–94)	75 (48–108)	14	82 (60–122)	22	73 (48–95)
6–14 (*n* = 175)	44 (20–82)	63	54 (34–89)	112	38 (19–72)	67 (41–97)	44	70 (50–98)	131	65 (40–95)
15–17 (*n* = 53)	29 (10–82)	11	55 (27–89)	42	24 (10–61)	61 (39–99)	7	74 (44–102)	46	58 (39–92)
Total boys (*n* = 264)	43 (19–82)	92	56 (33–92)	172	35 (16–74)	67 (40–98)	65	73 (50–101)	199	65 (39–95)
Men										
18–29 (*n* = 118)	38 (17–73)	24	46 (32–77)	94	35 (16–70)	68 (31–105)	21	77 (49–102)	97	66 (30–109)
30–39 (*n* = 209)	37 (15–84)	51	53 (26–95)	158	33 (14–75)	66 (38–106)	44	75 (42–103)	165	64 (36–98)
40–49 (*n* = 271)	40 (18–83)	82	63 (32–91)	189	33 (17–66)	67 (38–106)	62	74 (41–121)	209	66 (36–101)
50–59 (*n* = 225)	47 (19–88)	71	65 (36–102)	154	40 (19–84)	70 (36–108)	65	72 (42–108)	160	69 (34–107)
60+ (*n* = 225)	56 (22–96)	94	67 (37–99)	131	41 (18–94)	73 (41–111)	87	79 (45–108)	138	69 (36–116)
Total men (*n* = 1048)	43 (18–87)	322	63 (32–97)	726	36 (16–80)	69 (37–105)	279	74 (41–109)	769	66 (36–102)
Girls										
2–5 (*n* = 39)	59 (20–101)	24	67 (41–101)	15	30 (18–83)	72 (50–106)	17	78 (58–106)	22	66 (49–94)
6–14 (*n* = 164)	43 (17–74)	52	52 (29–84)	112	39 (16–67)	67 (39–96)	55	68 (44–96)	109	65 (33–87)
15–17 (*n* = 60)	47 (19–84)	11	63 (48–112)	49	38 (18–84)	67 (39–117)	13	78 (33–121)	47	65 (40–108)
Total girls (*n* = 263)	45 (18–84)	87	57 (33–101)	176	38 (17–71)	68 (40–102)	85	72 (49–103)	178	65 (38–102)
Women										
18–29 (*n* = 233)	49 (20–102)	66	67 (38–121)	167	43 (17–88)	76 (43–122)	50	78 (43–122)	183	75 (44–122)
30–39 (*n* = 280)	46 (20–99)	118	66 (29–107)	162	35 (17–85)	69 (41–117)	99	76 (47–118)	181	65 (38–114)
40–49 (*n* = 384)	49 (19–94)	170	64 (33–103)	214	40 (16–82)	71 (37–109)	147	75 (42–109)	237	67 (35–110)
50–59 (*n* = 335)	63 (23–111)	159	73 (42–115)	176	50 (19–90)	77 (40–117)	139	82 (47–127)	196	74 (34–116)
60+ (*n* = 285)	71 (24–109)	165	81 (47–117)	120	51 (21–98)	81 (45–125)	146	86 (58–129)	139	73 (37–118)
Total women (*n* = 1517)	57 (21–103)	678	72 (34–114)	839	43 (18–88)	74 (40–118)	581	81 (47–123)	936	70 (37–116)

^a^ SU, supplement users, i.e., those with an intake of vitamin D from supplements equal to or greater than 2 μg/day; ^b^ NSU, non-supplement users.

**Table 2 nutrients-10-01801-t002:** Percentage distribution of 25(OH)D) concentrations (nmol/L) among men/boys and women/girls within age groups and for spring and autumn.

	25(OH)D (nmol/L) Range in Spring Percentage Distribution within Each Age Group					25(OH)D (nmol/L) Range in Autumn Percentage Distribution within Each Age Group				
Age in Years by Sex (*n*)										
	<12.5	12.5–<25	25–<50	50–<75	>75	<12.5	12.5–<25	25–<50	50–<75	>75
Boys										
2–5 (*n* = 36)	0	11	33	36	20	0	0	6	44	50
6–14 (*n* = 175)	0	14	46	34	6	0	1	14	57	28
15–17 (*n* = 53)	7	36	38	11	8	0	0	26	53	21
Total boys (*n* =264)	2	18	42	30	8	0	0	16	54	30
Men										
18–29 (*n* = 118)	0	22	46	28	4	0	1	23	37	39
30–39 (*n* = 209)	2	20	50	18	10	0	1	20	47	32
40–49 (*n* = 271)	1	16	51	24	8	0	1	19	46	34
50–59 (*n* = 225)	2	8	45	32	13	0	2	14	48	36
60+ (*n* = 225)	1	7	35	37	20	0	0	12	45	43
Total men (*n* = 1048)	1	14	45	28	12	0	1	17	45	37
Girls										
2–5 (*n* = 39)	0	5	36	33	26	0	0	5	54	41
6–14 (*n* = 164)	1	9	56	29	5	0	1	13	58	28
15–17 (*n* = 60)	2	10	43	35	10	0	0	17	51	32
Total girls (*n* = 263)	1	9	50	31	9	0	1	12	56	31
Women										
18–29 (*n* = 233)	1	11	40	30	18	0	1	11	37	51
30–39 (*n* = 280)	1	12	42	25	20	0	1	14	49	36
40–49 (*n* = 384)	1	10	40	31	18	0	0	15	45	40
50–59 (*n* = 335)	0	5	27	37	31	0	1	11	34	54
60+ (*n* = 285)	0	6	18	31	45	0	0	8	33	59
Total women (*n* = 1517)	1	9	33	31	26	0	1	12	39	48

**Table 3 nutrients-10-01801-t003:** Vitamin D status in autumn and spring cross-tabled among adults and children and restricted to non-supplement users for both age groupings.

	All			Non-Supplement Users ^a^		
	Spring			Spring		
	<25 nmol/L	25–50 nmol/L	>50 nmol/L	<25 nmol/L	25–50 nmol/L	>50 nmol/L
**Autumn**	Adults (*n* = 2565; 18–69 years)			Adults (*n* = 1390)		
<25 nmol/L	1%	0%	0%	1%	0%	0%
25–50 nmol/L	6%	7%	1%	10%	8%	1%
>50 nmol/L	5%	31%	49%	8%	43%	29%
**Autumn**	Children (*n* = 527; 2–17 years)			Children (*n* = 308)		
<25 nmol/L	0%	0%	0%	1%	0%	0%
25–50 nmol/L	8%	6%	1%	12%	6%	0%
>50 nmol/L	7%	40%	38%	10%	49%	22%

^a^ Non-supplement users are those with an intake of vitamin D from supplements below 2 μg/day in both spring and autumn.

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
