# Peer review of "Vitamin D Status and Seasonal Variation among Danish Children and Adults: A Descriptive Study"

_nutrients, 2018, doi:10.3390/nu10111801_

Round 1
Reviewer 1 Report
This is an interesting study; however I have a number of concerns:
1). The introduction is quite basic and needs more information
2). It states that 'those at risk of osteoporosis were excluded'. Please clarify the criteria
3). Were the characteristics of the excluded participants significantly different from those included in the study?
4). Young children were included in the analysis - does this mean that Epi-vitamin D was also measured? Please clarify
5). Please detail if DEQAS or any international standardisation was used in the measurement. What were the lab assay CVs?
6). The paper is severely limited by the lack of data on sunlight exposure, sun holiday use, sunscreen use and dietary intakes of vitamin D. How do the authors value the worth of the results and conclusion due to these major constraints? Again how can the authors reliably explain the variation in vitamin D status without these important factors
7). There is no data on medication use, BMI, socio-economic status which we know affects vitamin D status. Please comment
Author Response
Please see attached PDF file.

Reviewer 2 Report
Line 149 vitamin insufficient in Spring, please add vitamin D insufficient in spring
Author Response
We have added the missing word, thank you for bringing this to our attention. The manuscript has been thoroughly checked for any spelling or grammatical errors, and we hope that the language is now satisfactory.
Reviewer 3 Report
This study describes the vitamin D status and seasonal variation in Danish population. Age, sex
and use of supplements were taken into considerations in evaluation of seasonal variation is serum vitamin D concentration.
Have the authors considered BMI of the patients? There is consistent association in the published literature between the BMI and lower serum 25 hydroxy vitamin D concentrations.
Author Response
Thank you for your comment. We agree that there is ample evidence linking BMI to lower serum 25(OH)D concentrations; however, this paper and study was meant to serve as a descriptive snapshot of the population – what does vitamin D status look like among the general Danish population if sampled randomly, in both children and adults, in those age groups where recommendations for supplement use does not exist. The paper was not meant to be analytical or to describe how vitamin D is associated with other factors (e.g. BMI). We hope that our explanation of the rationale of the paper is satisfactory.
Reviewer 4 Report
Review of Nutrients-2018-388141
Summary. This MS reports an observational study aimed at examining vitamin D status [serum 25(OH)D concentration, measured by LCMS, for seasonal variation in a sample of the general Danish population of children and adults under 70 years old, allowing for supplement usage and reports that seasonal variation and effects of supplementation were significant so that single 25(OH)D measurements are inadequate for assessing population D status.
Comments.
1.the study was limited to indigenous white Caucasians whilst poor D status is usually a greater problem in other ethnic groups, even when recommended to take supplementation, as in Denmark, so that a parallel study in such subjects is needed for formulation of public policy on vitamin D [and if in progress or nearing publication this should be mentioned]. For example, food fortificatio9n can be much more effective than simply recommending supplement usage. [e.g., as has been shown over recent years in Finland].
2.Despite the reference to an earlier paper describing how subjects were recruited, readers do need to be told, in brief, how this was done and to be told whether or not those recruited differed demographically from those not volunteering since, for example, those with higher education and/or higher incomes are often more likely to volunteer in such studies and also to maintain higher and more regular rates of vitamin D3 supplementation. These comparisons should also be reported fir those volunteers agreeing to provide monthly blood samples for D status compared to those not willing to do this.
3.The assay results reported are for the total of the 25(OH)D2 and D3 metabolites, but it would be of considerable interest to know whether the D2 metabolite is present in significant concentrations in significant numbers of people or not, especially if any Danish supplements provide intact vitamin D 2 rather than the usually preferable intact vitamin D3, or fungi/dark chocolate are much eaten.
4.Vitamin D is inactive and serum contents were not reported, therefore the term vitamin D really should not be used for reporting or discussing the assays of 25(OH)D, a point that should be checked throughout the MS for accuracy; in addition, the 25(OH)D assays, like any other measurement of compounds in blood or components if blood, are made as concentrations, so that this term should be used for reporting or discussing these assay data, reserving the term ‘level[s]’ for discussion of cut-offs, throughout the text. This has been done in general but should be checked [e.g., line 114].
5.Were there any examples of more than one family member being recruited to the study which could skew the findings, if so, does exclusion of such subjects after the findings at all?
6.figure 2 plots monthly data for serum 25(OH)D but the Legend is unclear as to which data plots are of means and which are of medians, please clarify. In addition, the plots show the mean/medians for all subjects bled monthly compared with those subjects not taking supplement, whereas, as a reader I would have liked to see the data for those taking vs. those not taking supplements – maybe this could be dealt with by adding the plot for the monthly mean/medians of the findings for 25(OH)D in the non-supplement takers?
7.The term ‘problematic’ is used in a few places, no doubt to imply that the use of the data referred to would be unreliable, but readers would, I suspect, feel it helpful to have the specific difficulty posed by that data spelt out rather more directly.
8.line 266, I think you mean implies, not entails.
9.line 267, I suggest that ‘clarified’, would be more accurate than ‘discussed’.
Author Response
Please see attached PDF file.

Round 2
Reviewer 1 Report
The authors have responded to all the reviewer comments